# Essential Oils Combined with Vitamin D3 or with Probiotic as an Alternative to the Ionophore Monensin Supplemented in High-Energy Diets for Lambs Long-Term Finished under Subtropical Climate

**DOI:** 10.3390/ani13152430

**Published:** 2023-07-27

**Authors:** Lucía de G. Escobedo-Gallegos, Alfredo Estrada-Angulo, Beatriz I. Castro-Pérez, Jesús D. Urías-Estrada, Elizabeth Calderón-Garay, Laura Ramírez-Santiago, Yissel S. Valdés-García, Alberto Barreras, Richard A. Zinn, Alejandro Plascencia

**Affiliations:** 1Faculty of Veterinary Medicine and Zootechnics, Autonomous University of Sinaloa, Culiacan 80260, Mexico; lucia.escobedo@uabc.edu.mx (L.d.G.E.-G.); alfred_vet@hotmail.com (A.E.-A.); laisa_29@hotmail.com (B.I.C.-P.); uriasestrada_jd@hotmail.com (J.D.U.-E.); elizabeth.calderon.fmvz@uas.edu.mx (E.C.-G.); laura.ramirez.fmvz@uas.edu.mx (L.R.-S.); 2Veterinary Science Research Institute, Autonomous University of Baja California, Mexicali 21100, Mexico; yissel.valdes@uabc.edu.mx (Y.S.V.-G.); beto_barreras@yahoo.com (A.B.); 3Animal Science Department, University of California, Davis, CA 95616, USA; razinn@ucdavis.edu

**Keywords:** natural additives, monensin, lambs, growth-performance, energetics, carcass

## Abstract

**Simple Summary:**

Natural feed additives have become a potential alternative to antibiotics used in ruminant diets as growth promoters. At present, there is interest in the search for combinations between natural additives that increase their effectiveness. In this sense, combining essential oils with other natural substances could have complementary effects that potentiate positive responses in performance and health in livestock. In this trial, compared to non-supplemented lambs or lambs supplemented with the antibiotic-ionophore monensin, the combination of essential oils with bacillus subtilis improved weight gain, whereas the combination of essential oils with vitamin D3 improved weight gain, gain efficiency, and some carcass traits. Those combinations showed a better response than antibiotic monensin to alleviate the adverse effects of a high-ambient temperature environment on growth performance or efficiency in feedlot lambs.

**Abstract:**

Supplementation with natural additives such as essential oils (EO) or probiotics has resulted in comparable growth performance to that of supplemental monensin in fattening lambs in hot environments. Supra-supplementation levels of vitamin D3 improved the carcass weight and dressing percentage of steers fattened under tropical conditions. We hypothesized that certain combinations of these natural additives could be complementary. For this reason, a feeding trial was carried out using 48 Pelibuey × Katahdin non-castrated male lambs (107 ± 14 d age; 17.9 ± 2.51 kg LW). Lambs were fed an 88:12 concentrate to forage ratio basal diet supplemented (dry matter basis, DMI) with: (1) no additive (CON); (2) 28 mg monensin/kg diet (MON); (3) 150 mg of essential oils containing a combination of thymol, eugenol, vanillin, guaiac, and limonene plus 0.12 mg vitamin D3 (EO + D3)/kg diet; and (4) 300 mg of essential oils containing a combination of carvacrol and cynamaldehyde plus 2 g probiotic (2.2 × 10^8^ CFU of bacillus subtilis/kg diet, EO + BS). Lambs were grouped by initial weight and assigned within six weight groupings to 24 pens (2 lambs/pen, 6 replicas per treatment) in a randomized complete block design. The experiment lasted 121 days. Daily maximal THI exceeded the 80 “danger or “emergency” range for 119 days of the 121 days of the trial. Lambs supplemented with MON had similar DMI, growth performance, and dietary energetics to those of CON lambs. Lambs supplemented with EO + BS had a greater (9.2%, *p* ≤ 0.05) average daily gain (ADG) than the CON and MON groups due to enhanced (10.2%, *p* ≤ 0.05) dry matter intake. Thus, gain efficiency (GF) and estimated dietary energy were similar for CON, MON, and EO + BS. Lambs receiving EO + D3 had similar (0.254 vs. 0.262 kg/d) ADG but a lower DMI (8%, *p* < 0.05) compared with EO + BS lambs. Consequently, GF and estimated dietary net energy were greater (4.9 and 3.7%, respectively; *p* ≤ 0.05) for EO + D3 lambs. Even when ambient heat load was elevated, the efficiency of utilization of dietary energy (observed-to-expected dietary net energy) was close to 1.00 (0.992) expected for EO + D3 lambs. In contrast, efficiency of energy utilization was depressed by −4.4% for lambs on the other treatments. Compared with the other treatments, lambs receiving EO + D3 had greater longissimus muscle area (5.6%, *p* < 0.05) and lower kidney pelvic fat (21.8%, *p* ≤ 0.05). There were no treatment effects on shoulder tissue composition or whole cuts (expressed as % of cold carcass weight). Compared to CON, lambs that were fed with natural additives showed 3.5% lower (*p* ≤ 0.05) intestine mass. All supplemental additives decreased visceral fat mass, which was minimal with EO + D3 treatment. Combinations of essential oils with vitamins or probiotics were superior to antibiotic monensin in finishing diets for feedlot lambs. Combining EO with probiotics promoted DM intake and gain but not gain efficiency, while combining EO with vitamin D3 supra-supplementation increased dietary energy efficiency and improved some carcass characteristics in lambs fattening under high ambient heat loads.

## 1. Introduction

Animal meat production under conditions of elevated ambient heat load represents a challenge for sustaining efficient productivity. Several strategies have been developed to mitigate the negative impact of high ambient heat loads [1]. Among these, some feed additives have shown promising results. The antibiotic-ionophore monensin is widely fed in countries where it is approved (i.e., Canada, USA, Mexico, Brazil, New Zealand, Argentina, Chile, and South Africa, among others) [2]. Monensin may reduce the energy requirement for maintenance in cattle under high ambient loads, increasing gain efficiency [3,4]. However, as a result of the potential risk of the escalation of antibiotic-resistant bacteria, antibiotic-free meat production is an increasingly evident concern in the livestock industry. In the last decades, several reports indicate that natural feed additives have become a potential alternative to antibiotics used as growth promoters (AGP) in livestock [5]. At present, pharmaceutical companies have been creating more stable products based on modified live strains as well as new products consisting of standardized and specific blends of essential oils. This new product requires constant evaluation under varied rearing conditions. In ruminants, supplementation with standardized essential oils (EO) resulted in comparable growth performance to that of supplemental monensin in both temperate and hot environments [6,7]. Probiotics can also alleviate the deleterious effects of high ambient temperatures on feedlot cattle performance [8] and finishing lambs [9]. It is well known that combining natural additives from different natures can potentiate, through synergistic or complementary effects, the magnitude of positive responses on animal health and productivity [10,11]. In this regard, combining essential oils (EO) with exogenous enzymes has shown complementary effects on weight gain and gain efficiency in feedlot cattle and feedlot lambs [12,13]. In the same line, compared with monensin alone or the combination monensin plus virginiamycin, combining a standardized blend of EO (thymol, eugenol, vanillin, guaiac, and limonene) with 25-hydroxyvitamin D3 enhanced feedlot cattle growth performance during periods of high ambient heat load [14]. To our knowledge, there are no reported studies evaluating the effects of combining EO with probiotics on the growth performance of ruminants. However, combinations of carvacrol with lactobacillus subtilis have shown positive effects on health and productivity in pigs and broilers [15,16]. The objective of this experiment was to compare the effects of combinations of two standardized sources of EO, one composed of a blend containing thymol, eugenol, vanillin, guaiac, and limonene combined with 25-hydroxyvitamin D3, as well as a combination of EO containing carvacrol and cynamaldehyde combined with the probiotic bacillus subtilis vs. monensin, on growth performance, dietary energy utilization, carcass traits, whole cuts, and visceral in feedlot lambs finished under subtropical climatic conditions.

## 2. Materials and Methods

### 2.1. Location Where the Study Was Performed

The experiment was conducted at the Universidad Autónoma de Sinaloa Feedlot Lamb Research Unit, located in Culiacán, México (24°46′13″ N and 107°21′14″ W). Culiacán is about 55 m above sea level and has a tropical climate. During the course of the experiment (May to September 2022), ambient air temperature averaged 34.2 °C (minimum and maximum of 31.7 °C and 36.0 °C, respectively), and relative humidity averaged 44.6% (minimum and maximum of 40.1% and 53.7%, respectively). All animal management procedures were conducted within the guidelines of federally and locally approved techniques for animal use and care [17] and approved by the Ethics Committee of the Faculty of Veterinary Medicine and Zootechnics of the Autonomous University of Sinaloa (Protocol #05202022).

### 2.2. Climatic Variables and Temperature Humidity Index (THI) Estimation

Climatic variables (ambient temperature and relative humidity) were obtained every hour from two on-site weather stations (Thermo-hygrometer Avaly, Mod. DTH880, Mofeg S.A., Zapopan, Jalisco, Mexico). The temperature humidity index (THI) was calculated using the following formula: THI = 0.81 × T + (RH/100) × (T − 14.40) + 46.40, where T = temperature expressed in Celsius grade and RH = relative humidity [18].

### 2.3. Animals, Treatments, and Experiment Design

With the aim of evaluating the treatments, 32 days before the start of the trial, a total of 55 Pelibuey × Katahdin crossbred intact male lambs were received at the research facility. Upon arrival, lambs were treated for parasites (7.5 mg/kg LW; Closantel Panavet 15%, Panamericana Veterinaria de México City, México), injected with 2 mL vitamin A (500,000 UI, 75,000 IU vitamin D3, and 50 IU vitamin E; Synt-ADE^®^, Zoetis México, México City, Mexico), and vaccinated for Mannheimia haemolytica (One Shot Ultra, Zoetis México, México City, Mexico). Lambs were adapted to the basal diet (without additives; Table 1) during the 32-day period before the initiation of the experiment. At the initiation of the study, lambs were individually weighed before the morning meal (electronic scale; TORREY TIL/S: 107 2691, TOR REY Electronics Inc., Houston, TX, USA). From the original group of 55 lambs, 48 lambs (107 ± 14 d age; 17.9 ± 2.51 kg BW) were selected, based on the uniformity of weight and general condition, for use in the experiment. The selected lambs were blocked by weight into six weight groupings and assigned within each weight grouping to 24 pens (two lambs/pen and six replicas per treatment). Pen dimensions were 6 m^2^ with overhead shade, automatic waterers, and 1 m fence-line feed bunks. A cracked corn-based total mixed ration was used as a basal diet (corn cracked for a final density of approximately 0.52 kg/L) in which ground sudangrass hay was used as a forage source. Sudangrass hay was ground in a hammer mill (Azteca 20, Molinos Azteca, Guadalajara, México) with a 3.81-cm screen before incorporation into a total mixed ration. The ingredients and chemical composition of the basal diet are shown in Table 1. Treatments consisted of the basal diet supplemented with: (1) no additives (CON); (2) 28 mg monensin (Elanco Animal Health, Indianapolis, IN, USA)/kg diet (MON); (3) 150 mg essential oils based on thymol, eugenol, vanillin, guaiac, and limonene plus 0.12 mg vitamin D3 (DSM Nutritional Products, Basel, Switzerland)/kg diet (EO + D3) and (4) 300 mg essential oils based on carvacrol and cynamaldehyde plus 2 g probiotic (2.2 × 10^8^ CFU/g bacillus subtilis; Kemin Industries, Des Moines, IA, USA)/kg diet (EO + BS). The recommended daily dose of MON for increased feed efficiency in finishing lambs is between 20 and 40 mg MON [19]. The doses used for the additives in Kemin Industries were according to label fact sheets. The dose used for the combination EO plus D3 was the same as used in previous reports where positive growth performance responses were detected [13,14]. Complete mixed diets were prepared using a 2.5 m^3^ capacity paddle mixer (model 30910-7, Coyoacán, México) as follows: (1) The grain portion of the diet was added to the mixer; (2) the dry supplement (mineral supplement and zeolite) was added to the mixer; (3) we allowed the feed to mix for a minimum of 1 min; (4) then, ground forage hay was added; (5) the fat component of the diet was added after the forage hay; (6) as the last step, molasses was added to the mixer and allowed to mix for approximately 7 min. To avoid cross-contamination between treatments, the mixer was thoroughly cleaned between each batch. To ensure additive consumption, the total daily dosage per lamb was mixed into 300 g of basal diet provided in the morning feeding (all lambs were fed the basal control diet in the afternoon feeding). Thus, lambs were provided fresh feed twice daily at 0800 and 1400 h. Whereas the amount of feed provided in the morning feeding was constant, the feed offered in the afternoon feeding was adjusted daily, allowing for a maximal feed residual ~100 g/pen. Residual feed was collected daily between 0740 and 0750 h each morning and weighed. The adjustments to either increase or decrease daily feed delivery were provided in the afternoon feeding. Lambs were weighed just prior to the morning feeding on days 1 and 121 (the final day) of the experimental period. Live weights (LW) on day 1 were converted to shrunk body weight (SBW) by multiplying LW by 0.96 to adjust for the gastrointestinal fill [20]. All lambs fasted for 18 h before recording the final LW.

### 2.4. Sample Analysis

Feed samples were collected from each batch. Feed refusal was collected daily and composited weekly for DM analysis (oven drying at 105 °C until no further weight loss; method 930.15) [21]. Feed samples were subjected to the following analyses: DM (oven drying at 105 °C until no further weight loss; method 930.15) and CP (N × 6.25; method 984.13) according to AOAC [21]. Neutral detergent fiber (NDF) was determined following procedures described by Van Soest et al. (corrected for NDF-ash, incorporating heat-stable α-amylase using Ankom Technology, Macedon, NY, USA) [22].

### 2.5. Calculations

Estimates of ADG and dietary net energy are based on initial SBW and final (d 121) fasted SBW. The average daily gain was computed by subtracting the initial SBW from final SBW and dividing the result by the number of days on feed. Feed efficiency was computed as ADG/average DMI observed during the 121 days of the experiment. One approach for evaluation of the efficiency of dietary energy utilization in growth-performance trials is the ratio of observed-to-expected DMI and observed-to-expected dietary NE. Based on estimated diet NE concentration and measures of growth performance, there is an expected energy intake. This estimation of expected DMI is performed based on observed ADG, average SBW, and NE values of the diet (Table 1): expected DMI, kg/d = (EM/2.04) + (EG/1.40), where EM (energy required for maintenance, Mcal/d) = 0.056 × SBW^0.75^, EG (energy gain, Mcal/d) = 0.276 × ADG × SBW^0.75^, and 2.04 and 1.40 are the NE_m_ and NE_g_ values contained in the basal diet (Table 1). Those values were calculated based on the ingredient composition [23] in the basal diet (Table 1). The coefficient (0.276) was taken from NRC [24], assuming a mature weight of 113 kg for Pelibuey × Katahdin male lambs [25]. The observed dietary net energy was calculated using EM and EG values and the DMI observed during the experiment by means of the quadratic formula:$$x=\frac{-b\pm \sqrt{b^{2}-4ac}}{2c}$$
where *x* = observed dietary NE, Mcal/kg, *a* = −0.41EM, *b* = 0.877 EM + 0.41 DMI + EG, and *c* = −0.877 DMI [26].

**Table 1 animals-13-02430-t001:** Composition of dietary treatments offered to lambs.

	Treatments ^§^			
Item	CON	MON	EO + D3	EO + BS
Ingredient composition, % DM basis				
Dry-rolled corn	55.00	55.00	55.00	55.00
Sudangrass hay	11.50	11.50	11.50	11.50
Soybean meal	15.00	15.00	15.00	15.00
Monensin	0	+++	0	0
Essential oils plus 25-Hydroxi-D3	0	0	+++	0
Essential oils plus probiotics	0	0	0	+++
Molasses cane	10.00	10.00	10.00	10.00
Zeolite	2.00	2.00	2.00	2.00
Tallow	4.00	4.00	4.00	4.00
Mineral-protein supplement *	2.50	2.50	2.50	2.50
Chemical composition (%DM basis) ^‡^				
Dry matter	88.22	88.22	88.22	88.22
Neutral detergent fiber	15.11	15.11	15.11	15.11
Crude protein	15.43	15.43	15.46	15.43
Ether extract	6.10	6.10	6.10	6.10
Calculated net energy (Mcal/kg) ^⁋^				
Maintenance	2.04	2.04	2.04	2.04
Gain	1.40	1.40	1.40	1.40

The symbol +++ = included in diet. ^§^ MON = Monensin 28 mg/kg diet DM (Rumensin 90^®^, Elanco Animal Health, Indianapolis, IN, USA); EO + D3 = standardized source of a mixture of essential oils 150 mg/kg diet DM (CRINA; DSM Nutritional Products, Basel, Switzerland) plus 0.12 mg/kg diet DM of 25-hydroxy-vitamin-D_3_ (HyD_3;_ DSM Nutritional Products, Basel, Switzerland); EO + BS = 300 mg essential oils (PrintArome, NOREL Nutritición Animal, Queretaro, Mexico) plus 2 g/kg diet of a product containing bacillus subtilis 2.2 × 10^8^ CFU (CLOSTAT dry, Kemin Industries, Des Moines, IA, USA). * Mineral premix contains: Premix contained: Limestone, 50%; urea, 20%; NaCl, 15%, MgO, 5%, phosphate rock, 9.06%; CoSO_4_, 0.01%; CuSO_4_, 0.14%; FeSO_4_, 0.47%; ZnO, 0.16%; MnSO_4_, 0.14%; KI, 0.008%. ^‡^ Based on tabular values for individual feed ingredients [23], with the exception of CP and NDF, which were determined in our laboratory [21,22]. ^⁋^ Based on tabular energy values for individual feed ingredients informed by NRC [21].

### 2.6. Carcass Characteristics, Whole Cuts, and Tissue Shoulder Composition

All the lambs were slaughtered on the same day. After humanitarian sacrifice [17], lambs were skinned, and the gastrointestinal organs were separated and weighed. After carcasses (with kidneys and internal fat included) were chilled in a cooler at −2 to 1 °C for 24 h, the following measurements were obtained: (1) body wall thickness (at a point between the 12th and 13th rib, five inches from the midline of the carcass); (2) fat thickness perpendicular to the *m. longissimus thoracis* (LM), measured over the center of the ribeye between the 12th and 13th rib; (3) LM surface area, measured using a grid reading of the cross sectional area of the ribeye between the 12th and 13th rib; and (4) kidney, pelvic, and heart fat (KPH). The KPH was manually removed from the carcass, weighed, and reported as a percentage of the cold carcass weight (CCW) [27]. Each carcass was split into two halves. The left side was fabricated into wholesale cuts without trimming, according to the North American Meat Processors Association guidelines [28]. Rack, breast, shoulder, and foreshank were obtained from the foresaddle, and the loins, flank, and leg from the hindsaddle. The weight of each cut was subsequently recorded. The tissue composition of the shoulder was assessed using physical dissection using the procedure described by Luaces et al. [29].

### 2.7. Visceral Mass Data

Components of the gastro-intestinal tract (GIT), including the tongue, esophagus, stomach (rumen, reticulum, omasum, and abomasum), pancreas, liver, gallbladder, small intestine (duodenum, jejunum, and ileum), and large intestine (caecum, colon, and rectum), were removed and weighed. The GIT was then washed, drained, and weighed to obtain empty weights. The difference between full and washed digesta-free GIT was subtracted from the SBW to determine the empty body weight (EBW). All tissue weights are reported on a fresh tissue basis. Organ mass is expressed as grams of fresh tissue per kilogram of final EBW, where final EBW represents the final full live weight minus the total digesta weight. The full visceral mass was calculated by the summation of all visceral components (stomach complex + small intestine + large intestine + liver + lungs + heart), including digesta. The stomach complex was calculated as the digesta-free sum of the weights of the rumen, reticulum, omasum, and abomasum.

### 2.8. Statistical Analysis

Growth performance data (gain, gain efficiency, and dietary energetics), DM intake, and carcass data were analyzed as a randomized complete block design, with the pen as the experimental unit, using the MIXED procedures of SAS software 9.3 [30], with treatment and block as fixed effects and the experimental unit within treatment as a random effect. Visceral organ mass data was analyzed using the MIXED procedures of SAS software [30], with treatment and pen as fixed effects and interaction between treatment and pen and individual carcasses within pen by treatment subclasses as random effects. Treatment means were separated using the “honestly significant difference test” (Tukey’s HSD test). In all cases, the least squares mean and standard error are reported, and contrasts are considered significant when the *p* value ≤ 0.05.

## 3. Results

### 3.1. Ambient Temperature

The temperature and relative humidity during the experiment are presented in Table 2. The average minimum and maximum estimated THI were 74.99 and 87.90, respectively. Daily maximal THI exceeded 80 for 119 d of the 121 days of the trial, corresponding to “danger” conditions according to the code of Mader et al. [31]. The number of hours per day that THI exceeded 80 was 6.2 ± 1.02. Accordingly, lambs were under high ambient load conditions throughout the experiment.

### 3.2. Additives Intake

Based on measures of feed intake, average daily intakes of additives were: MON, 27.58 mg (equivalent to 0.88 mg/kg LW); EO + D3, 4.8 and 0.004 mg/kg LW, for EO and D3, respectively; EO + BS, 10.15 mg and 0.07 g/kg LW, for EO and BS, respectively. The recommended daily dose of MON for increased feed efficiency in finishing lambs is between 20 and 40 mg MON [19], while the doses received in the combinations EO + D3 and EO + BS were within doses that have been reported to have positive effects on growth performance and carcass. Therefore, the final doses ingested in the current study should not represent a limiting factor for the responses evaluated.

### 3.3. Growth Performance and Dietary Energy

Treatment effects on growth performance and dietary energetics are shown in Table 3. Lambs that were supplemented with MON showed a comparable DMI (averaging 1.001 kg DM/d) to CON lambs. In a similar manner, lambs supplemented with MOM had similar growth performance and dietary energetics to CON lambs. Lambs supplemented with EO + BS had a greater (9.2%, *p* ≤ 0.05) ADG than CON and MON. However, the increased ADG is attributable to an increased DMI (10.2%, *p* ≤ 0.05). Gain efficiency (GF) and estimated dietary net energy were similar (*p* > 0.28) between CON, MON, and EO + BS. Lambs receiving EO + D3 had similar ADG (0.254 vs. 0.262 kg/d, *p* = 0.39), but a lower DMI (8%, *p* = 0.04) than EO + BS lambs. Accordingly, GF and estimated dietary net energy were greater (4.9 and 3.7%, respectively; *p* < 0.01) for EO + D3 vs. EO + BS treatments. Notwithstanding the elevated ambient heat load, the efficiency of utilization of dietary energy (observed-to-expected dietary net energy) was close to 1.00 (0.992) for EO + D3 lambs. Whereas, for the other three treatments (CON, MON, and EO + BS), the estimated efficiency of dietary energy utilization averaged −4.4% less than expected.

### 3.4. Carcass and Visceral Mass

Treatment effects on carcasses and whole cuts are presented in Table 4. Carcass traits were very similar (*p* > 0.05) between CON and MON lambs. Compared to the other treatments, lambs receiving EO + BS showed a heavier carcass weight (3.6%, *p* < 0.05), without differences with CON and MON lambs in the rest of the carcass traits. Compared to the other treatments, lambs receiving EO + D3 had greater longissimus muscle area (5.6%, *p* < 0.05) and less kidney pelvic fat (21.8%, *p* ≤ 0.05). There were no treatment effects on shoulder tissue composition (expressed as a percentage) or in whole cuts (expressed as a percentage of cold carcass weight). Compared to CON, lambs that were fed with EO + D3 and EO + BS showed less intestinal mass (5.8%, *p* ≤ 0.05), but were similar (*p* = 0.37) to MON. Compared to the CON group, all supplemental additives decreased (*p*) visceral fat mass (Table 5).

## 4. Discussion

A high ambient heat load does not necessarily cause heat stress. Numerous studies (long-term experiments) did not reveal changes in “stress parameters” (rectal temperature, blood metabolites, breath rate, etc.). Under such conditions (THI > 79), animals adapt by reducing the heat load associated with energy intake [32,33]. But “adaptation physiology” can affect the energy utilization of the diet and growth performance. An objective of this experiment was to evaluate the effect of the additives tested on intake patterns and estimates of dietary energy utilization, as well as carcass traits, in adapted feedlot lambs that are finished in a subtropical climate during the summer season.

Due to climatic conditions in subtropical regions, challenges exist when sustaining efficient productivity in livestock systems. Although, the ambient heat load has a lesser impact on DMI in hairy lambs and their crosses than non-adapted breeds [34,35], the main challenge faced in the finishing phase is maintaining an ADG that is consistent with genetic potential in periods of more extreme ambient heat loads. According to NRC [36], the predicted feed intake of sheep consuming milled diets containing 2.04 Mcal/kg NE_m_ is 83.50 g DM/kg LW^0.75^. This is in agreement with the DM intake of 81 g DM/kg LW^0.75^ registered at this Research Center for lambs of similar breeding, average weight, and dietary energy concentration used in this experiment but fattened in favorable climatic conditions (THI < 76) [13,37,38]. Applying the standards for dietary NE utilization [23], the average DMI for the CON and EO + D3 observed in the present experiment was 7.8% less than anticipated. Whereas, DMI for EO + BS lambs closely matched expectations (1.123/1.119 kg DM/d). The reduction of DMI observed in the control group and EO + D3 treatments is in close agreement with reductions of 8.3% observed by Macías-Cruz et al. [39] in a comparison of GF of hairy lambs during the summer vs. spring season under semi-arid environments. Because of the wide diversity of form and composition of EO, their effects on DMI likewise vary [40]. In this sense, compared to non-supplemented treatments, the blend of essential oils (thymol, eugenol, vanillin, guaiac, and limonene) used alone or in the combination EO + D3 showed a similar DMI in previous studies when supplemented in feedlot cattle [41,42,43] and in lambs [13,44]. Similarly, a blend of carvacrol and cynamaldehyde or cynamaldehyde alone failed to increase DMI when supplemented in lambs at levels of up to 400 mg/kg diet DM [45,46] or in lactating cows [47].

The dry matter intake of CON and EO + D3 lambs was not different from that of MON. Characteristically, MON supplementation in feedlot cattle tends to reduce DMI [48]. However, in feedlot lambs, this effect has been less consistent. In some cases, MON supplementation decreased lamb DMI [49,50], whereas in other cases, no effect was detected [51,52,53]. Furthermore, when feedlot cattle were exposed to hot environments (THI reached a value of 79 or higher, similar to the prevailing THI in the present experiment), the difference in DMI between feedlot cattle supplemented with MON vs. non-supplemental control was not appreciable [4]. Decreased DMI is the primary basis for reduced ADG in animals exposed to high ambient heat loads. However, some of the reduction in ADG may be associated with decreased efficiency of energy utilization related to the additional energy cost of body heat dissipation [33,54].

In growing-finishing trials, efficiency of energy utilization can be assessed by comparing observed vs. expected dietary net energy based on measures of growth performance. Estimation of dietary NE based on measures of growth performance provides important insight into potential treatment (or environmental) effects on the efficiency of dietary energy utilization [55]. An observed-to-expected dietary NE ratio of 1.00 indicates that ADG is consistent with formulated dietary NE values based on tables of feedstuff standards [23] and observed DMI. A ratio that is greater than 1.00 indicates greater efficiency of dietary energy utilization, whereas a ratio that is lower than 1.00 indicates a lower than expected efficiency of energy utilization. Based on the above, control and MON treatment resulted in decreased ADG, but the reduction in ADG was only partially due to decreased DM intake. The ratio of observed-to-expected dietary NE was 0.960, a 4% decrease in efficiency in dietary energy utilization. In the case of the EO + BS-supplemented lambs, both ADG and DMI were enhanced. However, DMI was 5% greater than expected based on observed ADG (the ratio of observed-to-expected dietary NE was 0.95), indicative of decreased efficiency in dietary energy utilization. Therefore, it appears that this particular combination (thymol and carvacrol with BS) promotes DM intake and weight gain, but without improvement in the efficiency of dietary energy utilization. Likewise, Tan et al. [15] observed that supplementation of weaned pigs reared under high ambient heat with Bacillus subtilis in combination with a blend of essential oils composed of thymol and carvacrol increased ADG but did not improve gain efficiency when compared to non-supplemented pigs.

Decreases in the efficiency of energy utilization may be attributable to increased maintenance energy requirements associated with elevated THI. The magnitude of changes in maintenance requirements for lambs can be estimated as follows: Maintenance coefficient (MQ) = (NE_m_ × [DMI − {EG/NE_g_}])/SBW^0.75^, where NE_m_ corresponds to the NE values of the diet (Table 1) according to NRC [23] tables, EG is the energy requirement for gain, and DMI and SBW correspond to the general average values of DMI and SBW observed during the experiment. Accordingly, in non-supplemented lambs, elevated THI increased the maintenance coefficient by 13% above the specified standard, 0.056 Mcal/SBW^0.75^ [56]. The magnitude of this increase in the maintenance coefficient is in good agreement with previously reported values (8.0, 14.7, and 16.0%) for non-supplemented Pelibuey × Katahdin lambs that were fed similar diets and under similar environmental conditions [13,38,53]. And, is within the expected range of 7% to 25% greater maintenance requirements for heat-stressed cattle mentioned previously [35,56]. Applying the same equation to MON and EO + BS treatments, the estimated maintenance requirement increased by 11% and 16%, respectively. In contrast, the observed to-expected dietary NE for EO + D3 was 0.992. Thus, notwithstanding the high ambient heat load, the apparent maintenance energy requirement was not increased. The observed ADG was consistent with dietary energy density and the observed DMI. Likewise, Mendoza-Cortés et al. [14] observed that in feedlot bulls supplemented with EO + D3 (119 mg EO plus 0.12 mg 25-hydroxi-vitamin-D3/kg diet) during an 84-d period in which ambient THI averaged 82.67, the maintenance requirement was not appreciably affected. The basis for this apparent mitigating effect of EO + D3 on energy requirements during periods of high ambient heat load is uncertain, meriting further investigation.

Differences in treatment effects on the LM area are expected based on changes in carcass weight. Consistent with previous studies [44,57,58,59], treatments effects on carcass traits and whole cuts were not appreciable. Potential antimicrobial and anti-inflammatory effects of probiotics and EO may account for the observed reduction in intestinal mass [13,60]. According to Wang et al. [61], supplemental essential oils (primarily thymol) decreased poultry jejunal wall thickness, while Ghazanfari et al. [62] found that in poultry, supplementation with essential oil mixtures (mixtures of linalool, terpinene, and limonene) decreased small intestine wall thickness by 30% on average. However, further studies are needed to elucidate how prebiotics and essential oils affect gut epithelial integrity and immune function [63]. It had been reported that EO and probiotics decreased intestinal wall thickness in mammals by inhibiting proinflammatory factors and activating protective proteins [64,65,66]. However, it warrants future research attention regarding the effects of probiotics, essential oils, or their combinations on intestinal mass in ruminants fed high-energy diets.

## 5. Conclusions

Based on growth performance, dietary energetics, and some carcass traits, combinations of essential oils with vitamin D3 or probiotics were superior to monensin in finishing diets for feedlot lambs. Combining EO with probiotics promoted DM intake and gain, but not efficiency. Combining EO with vitamin D3 enhanced the efficiency of dietary energy utilization and improved some carcass characteristics in lambs. Supplemental natural additive combinations showed a better response than antibiotic monensin to alleviate the adverse effects of a high-ambient temperature environment on growth performance or efficiency in feedlot lambs. More research is needed regarding the effectiveness of natural additive combinations on specific production systems. The metabolic and physiological basis for the apparent mitigating effect of EO + D3 on energy requirements during periods of high ambient heat load is not fully understood, meriting further investigation.

## Figures and Tables

**Table 2 animals-13-02430-t002:** Ambient temperature (Ta), mean relative humidity (RH), and mean calculated temperature-humidity index (THI) ^a^ were registered during the experiment.

Week	Mean T_a_ (°C)	Min T_a_ (°C)	Max T_a_ (°C)	Mean RH (%)	Min RH (%)	Max RH (%)	Mean THI	Min THI	Max THI
1	28.7 ± 3.5	24.5 ± 5.5	33.1 ± 1.6	52.2 ± 12.8	38.8 ± 6.2	65.5 ± 19.4	77.7 ± 4.9	70.1 ± 6.4	85.3 ± 3.4
2	31.1 ± 3.4	27.6 ± 4.4	34.5 ± 2.1	52.3 ± 13.0	40.1 ± 8.4	64.6 ± 17.6	80.6 ± 4.0	74.0 ± 4.7	87.2 ± 3.4
3	30.3 ± 3.0	27.1 ± 3.7	33.4 ± 2.28	55.1 ± 11.2	44.3 ± 8.2	66.0 ± 14.2	79.9 ± 3.8	73.9 ± 4.3	86.0 ± 3.2
4	33.1 ± 3.3	29.8 ± 4.8	36.5 ± 1.8	45.4 ± 10.8	35.8 ± 6.7	54.9 ± 14.9	81.9 ± 4.0	76.0 ± 5.2	87.9 ± 2.7
5	32.6 ± 3.1	28.9± 4.1	36.4 ± 2.1	54.4 ± 11.1	41.3 ± 6.0	67.4 ± 16.3	83.2 ± 4.2	75.8 ± 4.7	90.6 ± 3.6
6	31.1 ± 3.2	28.1 ± 3.6	34.0 ± 2.8	58.4 ± 14.6	45.6 ± 12.6	71.3 ± 16.7	81.5 ± 3.5	75.2 ± 3.7	87.7 ± 3.4
7	32.1 ± 3.0	28.9 ± 3.3	35.2 ± 2.7	56.1 ± 12.7	43.8 ± 10.1	68.4 ± 15.4	82.5 ± 3.7	76.1 ± 3.4	89.0 ± 4.1
8	33.1 ± 3.0	29.8 ± 3.9	36.5 ± 2.0	49.6 ± 10.4	38.5 ± 6.0	60.7 ± 14.7	82.8 ± 3.5	76.3 ± 4.1	89.3 ± 2.8
9	33.1 ± 2.5	29.6 ± 3.0	36.5 ± 2.1	52.4 ± 9.7	40.4 ± 6.1	64.5 ± 13.3	83.3 ± 2.9	76.4 ± 3.1	90.2 ± 2.6
10	30.7 ± 3.0	26.9 ± 3.2	34.51 ± 2.8	58.3 ± 11.8	44.4 ± 10.1	72.2 ± 13.6	81.2 ± 3.7	73.7 ± 3.7	88.7 ± 3.6
11	30.8 ± 2.7	27.1 ± 3.3	34.5 ± 2.1	59.8 ± 9.8	45.9 ± 7.7	73.7 ± 11.9	81.6 ± 3.2	74.1 ± 3.7	89.1 ± 2.7
12	31.5 ± 3.2	28.6 ± 3.1	34.4 ± 3.3	57.5 ± 10.3	46.8 ± 7.6	68.2 ± 13.0	81.9 ± 5.2	76.2 ± 3.9	87.7 ± 6.5
13	29.8 ± 2.5	27.1 ± 2.8	32.5 ± 2.3	65.3 ± 10.8	53.2 ± 8.4	77.4 ± 13.2	80.8 ± 2.6	75.0 ± 2.8	86.6 ± 2.4
14	30.7 ± 2.6	27.9 ± 2.9	33.4 ± 2.3	64.4 ± 10.6	52.4 ± 8.8	76.4 ± 12.4	81.9 ± 2.9	76.1 ± 3.1	87.9 ± 2.7
15	28.9 ± 3.0	26.6 ± 2.1	31.3 ± 3.9	71.4 ± 13.4	62.0 ± 15.1	80.8 ± 11.7	80.2 ± 3.4	75.3 ± 1.7	85.1 ± 5.1
16	30.6 ± 3.1	27.8 ± 3.2	33.5 ± 2.95	60.5 ± 11.6	48.4 ± 8.5	72.5 ± 14.6	81.3 ± 4.2	75.3 ± 3.7	87.4 ± 4.6
17	31.9 ± 2.5	28.8 ± 3.4	34.9 ± 1.5	53.9 ± 11.7	40.3 ± 7.1	67.7 ± 16.2	82.1 ± 3.7	75.5 ± 4.0	88.6 ± 3.5
Avg	31.2 ± 3.0	27.9 ± 3.5	34.4 ± 3.4	56.9 ± 11.5	44.8 ± 8.4	68.9 ± 14.7	81.4 ± 3.7	74.9 ± 3.9	87.9 ± 3.5

^a^ THI = 0.81 × ambient temperature + [(relative humidity/100) × (ambient temperature- 14.4)] + 46.4. THI code (Normal THI < 74; Alert > 74–79; Danger 79–84; and Emergency > 84) [29]. The experiment was conducted at the Universidad Autónoma de Sinaloa Feedlot Lamb Research Unit, located in Culiacán, México (24°46′13″ N and 107°21′14″ W). Culiacán is about 55 m above sea level, and has a tropical climate. During the course of the experiment (May to September 2022), ambient air temperature averaged 34.2 °C (minimum and maximum of 31.7 °C and 36.0 °C, respectively), and relative humidity averaged 44.6% (minimum and maximum of 40.1% and 53.7%, respectively).

**Table 3 animals-13-02430-t003:** Effect of treatments on growth performance of finishing lambs.

	Treatments ^†^				
Item	Control	MON	EO + D3	EO + BS	SEM
Days on test	121	121	121	121	
Pen replicates	6	6	6	6	
Live weight, kg/d ^§^					
Initial	17.9	18.1	17.8	18.1	0.113
Final	45.8 ^a^	44.9 ^a^	47.1 ^ab^	48.3 ^b^	0.825
Average daily gain, kg/d	0.231 ^ab^	0.221 ^a^	0.242 ^b^	0.250 ^b^	0.007
Dry matter intake, kg/d	1.030 ^a^	0.985 ^a^	1.033 ^a^	1.123 ^b^	0.029
Gain to feed ratio, kg/kg	0.236 ^a^	0.237 ^a^	0.246 ^b^	0.234 ^a^	0.003
Diet net energy, Mcal/kg					
Maintenance	1.970 ^a^	1.987 ^a^	2.035 ^b^	1.961 ^a^	0.016
Gain	1.316 ^a^	1.332 ^a^	1.375 ^b^	1.310	0.014
Observed-to-expected diet NE					
Maintenance	0.966 ^a^	0.976 ^a^	0.999 ^b^	0.963 ^a^	0.008
Gain	0.942 ^a^	0.954 ^a^	0.985 ^b^	0.938 ^a^	0.010
Observed-to-expected DMI	1.046 ^a^	1.035 ^a^	1.001 ^b^	1.051 ^a^	0.009

^a,b^ Means in a row with different superscript letters highlight statistical differences (*p* < 0.05). ^†^ MON = Monensin 28 mg/kg diet DM (Rumensin 90^®^, Elanco Animal Health, Indianapolis, IN, USA); EO + D3 = standardized source of a mixture of essential oils 150 mg/kg diet DM (CRINA; DSM Nutritional Products, Basel, Switzerland) plus 0.12 mg/kg diet DM of 25-hydroxy-vitamin-D_3_ (HyD_3;_ DSM Nutritional Products, Basel, Switzerland); EO + BS = 300 mg essential oils (PrintArome, NOREL Nutritición Animal, Queretaro, Mexico) plus 2 g/kg diet of a product containing bacillus subtilis 2.2 × 10^8^ CFU (CLOSTAT dry, Kemin Industries, Des Moines, IA, USA). ^§^ Initial shrunk weight is the full live weight reduced by 4% to adjust for gastrointestinal fill, final weight was obtained fasted for 18 h before recording the final LW.

**Table 4 animals-13-02430-t004:** Effect of treatments on carcass characteristics of finishing lambs.

	Treatments ^†^				
Item	Control	MON	EO + D3	EO + BS	SEM
Hot carcass weight, kg	26.9 ^ab^	26.6 ^a^	27.7 ^ab^	28.1 ^b^	0.443
Dressing percentage	56.9	57.5	57.0	56.4	0.004
Cold carcass weight, kg	26.8 ^a^	26.5 ^a^	27.2 ^ab^	27.9 ^b^	0.440
LM area, cm^2^	16.7 ^ab^	15.6 ^a^	17.2 ^b^	16.5 ^a^	0.395
Fat thickness, cm ^§^	0.217	0.206	0.240	0.233	0.015
Kidney pelvic and heart fat, %	3.61 ^a^	3.43 ^a^	2.76 ^b^	3.55 ^a^	0.190
Carcass yield *	1.25	1.22	1.35	1.32	0.062
Shoulder composition, %					
Muscle	69.58	69.62	69.51	68.78	0.645
Fat	12.57	12.38	12.48	13.59	0.521
Muscle to fat ratio	5.54	5.62	5.57	5.06	0.228
Whole cuts (as percentage of CCW)					
Forequarter	40.59	39.83	41.10	40.67	0.692
Hindquarter	37.86	37.20	37.24	37.98	0.673
Neck	9.33	9.16	9.05	8.90	0.276
Shoulder IMPS206	7.85	7.83	8.08	7.89	0.277
Shoulder IMPS207	14.54	13.98	14.22	14.24	0.239
Rack IMPS204	6.85	6.63	6.97	6.99	0.126
Breast IMPS209	4.38	4.40	4.63	4.24	0.153
Ribs IMPS209A	6.90	6.94	6.89	7.26	0.167
Loin IMPS231	7.20	7.28	7.36	7.39	0.170
Flank IMPS232	6.08 ^ab^	5.74 ^a^	6.20 ^b^	6.38 ^b^	0.143
Leg IMPS233	24.54	24.16	24.62	23.55	0.497

^a,b^ Means a row with different superscripts differs (*p* < 0.05). ^†^ MON = Monensin 28 mg/kg diet DM (Rumensin 90^®^, Elanco Animal Health, Indianapolis, IN, USA); EO + D3 = standardized source of a mixture of essential oils 150 mg/kg diet DM (CRINA; DSM Nutritional Products, Basel, Switzerland) plus 0.12 mg/kg diet DM of 25-hydroxy-vitamin-D_3_ (HyD_3;_ DSM Nutritional Products, Basel, Switzerland); EO + BS = 300 mg essential oils (PrintArome, NOREL Nutritición Animal, Queretaro, Mexico) plus 2 g/kg diet of a product contained bacillus subtilis 2.2 × 10^8^ CFU (CLOSTAT dry, Kemin Industries, Des Moines, IA, USA). ^§^ Fat thickness over the center of the LM between the 12th and 13th ribs. * Carcass yield was estimated as (Fat thickness × 0.10) + 0.40.

**Table 5 animals-13-02430-t005:** Effect of treatments on the visceral mass of finishing lambs.

		Treatments ^†^			
Item	Control	MON	EO + D3	EO + BS	SEM
GIT fill, kg	3.44 ^ab^	2.97 ^a^	3.54 ^ab^	3.96 ^b^	0.317
Empty body weight, % of full weight	92.80	93.55	92.65	92.00	0.542
Full viscera, kg	8.00	7.51	8.08	8.54	0.276
Organs, g/kg of empty body weight					
Stomach complex	21.83	22.77	21.60	23.91	0.751
Intestines	46.02 ^a^	45.66 ^ab^	44.07 ^b^	44.76 ^b^	0.535
Liver/spleen	14.54	14.25	15.15	15.52	0.631
Heart/lungs	19.83	19.32	18.87	19.13	0.797
Kidney	2.69	2.77	2.79	2.87	0.119
Omental fat	35.17 ^a^	32.05 ^b^	26.58 ^c^	29.37 ^d^	0.869
Mesenteric fat	10.69 ^a^	9.46 ^ab^	8.29 ^b^	9.51 ^ab^	0.758
Visceral fat	45.86 ^a^	41.51 ^b^	34.87 ^c^	38.87 ^b^	1.346

^a,b,c^ Means a row with different superscripts differs (*p* < 0.05). ^†^ MON = Monensin 28 mg/kg diet DM (Rumensin 90^®^, Elanco Animal Health, Indianapolis, IN); EO + D3 = standardized source of a mixture of essential oils 150 mg/kg diet DM (CRINA; DSM Nutritional Products, Basel, Switzerland) plus 0.12 mg/kg diet DM of 25-hydroxy-vitamin-D_3_ (HyD_3;_ DSM Nutritional Products, Basel, Switzerland); EO + BS = 300 mg essential oils (PrintArome, NOREL Nutritición Animal, Queretaro, Mexico) plus 2 g/kg diet of a product containing bacillus subtilis 2.2 × 10^8^ CFU (CLOSTAT dry, Kemin Industries, Des Moines, IA, USA).

## Data Availability

Not applicable.

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
