# Peer review of "Essential Oils Combined with Vitamin D3 or with Probiotic as an Alternative to the Ionophore Monensin Supplemented in High-Energy Diets for Lambs Long-Term Finished under Subtropical Climate"

_animals, 2023, doi:10.3390/ani13152430_

Round 1
Reviewer 1 Report
Comments to the manuscript 2438306 “Natural Additives Combination as an Alternative to the Ionophore Monensin Supplemented in High-Energy Diets for Lambs Long-term Finished Under Subtropical Climate”, sent it to Animals. This study compared the effect of combination of essencial oils with vitamin D or combination of essencial oils with a bacteria (Bascilus subtilis) or monensin (Ionopher) on growth performance, dietary energy utilization, carcass traits, whole cuts, and visceral in feedlot lambs finished under subtropical climatic conditions (I assume that it was carry out during the summer season, authors didn´t mention it)
I have few comments and observations.
Line 114. Pens have 6 m2, change for Pens have 6 m2
Line 118. grasss, has an extra “s”
Line 184. m. longissimus thoracis …. What you mean?, what m means??
Table 2 is replicated as: Table 2. Ambient temperature (Ta)……… and Table 2. Effect of treatments on…..
Table 2. Ambient temperature (Ta), mean relative humidity (RH)……… has a trouble with the footnote.
Line 261….. heat. Loads, delete the dot
Line 280. The words “and abomasum.” are repeated, delete one.
In Table 3. Effect of treatments on…. The variables: Days on test and pen replicates, must be deleted due that them do not represent a mean and They were mentioned before in M&M.
Author Response
Response to REVIEWER 1
AU: We are grateful to reviewers for the time and effort in helping improve the quality of the manuscript. The observations were wise and timely which permit the improvement substantially the manuscript. We have addressed the concerns in our revised manuscript accordingly.
All changes and correction made are highlighted in yellow in the corrected version of the manuscript.
Responses
RW: Comments to the manuscript 2438306 “Natural Additives Combination as an Alternative to the Ionophore Monensin Supplemented in High-Energy Diets for Lambs Long-term Finished Under Subtropical Climate”, sent it to Animals. This study compared the effect of combination of essential oils with vitamin D or combination of essential oils with live bacteria (Bascilus subtilis) or monensin (Ionophore) on growth performance, dietary energy utilization, carcass traits, whole cuts, and visceral in feedlot lambs finished under subtropical climatic conditions (I assume that it was carry out during the summer season, authors didn´t mention it).
AU: The experiment was carried out during summer season (may to September) this information was incorporated in the correction version of the manuscript
I have few comments and observations.
RW: Line 114. Pens have 6 m2, change for Pens have 6 m2
AU: correction was done
RW: Line 118. grasss, has an extra “s”
AU: correction was done.
RW: Line 184. m. longissimus thoracis …. What you mean?, what m means??
AU: m. longissimus thoracis is the anatomy description of loin muscle. “m’ means “musculus”
RW: Table 2 is replicated as: Table 2. Ambient temperature (Ta)……… and Table 2. Effect of treatments on…..
AU: Thanks! Numbering of Tables were corrected.
RW: Table 2. Ambient temperature (Ta), mean relative humidity (RH)……… has a trouble with the footnote.
AU: Thanks! Footnote position is now corrected.
RW: Line 261….. heat. Loads, delete the dot
AU: Correction was made.
RW: Line 280. The words “and abomasum.” are repeated, delete one.
AU: Correction was made
Reviewer 2 Report
The conducted a research on the Ionophore Monensin Supplemented in High-Energy Diets for Lambs Long-term Finished Under Subtropical Climate. it looks good but it need improvement before any decision
simple summary is not good please revise
in abstract please add a line about the background of the study.
how many lambs were selected ? and how many lambs were selected in each group/
please ad the age of the lambs /
revise line L27-31
Revise L 31-33L 33-34 how?
results must be revised in abstract section
please revise keywords
Introduction is poor written and can not be acceptable in this condition
Give some title to first paragraph of the MM section
in section 2.1 add the figure of the temperature and humidity variation during the research trial please
please divided in to different sections
animal experiment design
sample analysis
can you a the figures of the cross breed >?
change L102-103 please rewrite
animal research design 103-136 must be revised confusing please rewrite and simplify it for proper understanding.
how you have prepared the Diet ? add the proper reference please
please check the diet composition carefully it looks not good revise it accordingly please
please carefully check the short forms which you have used in the manuscript
2.3 calculation please revise it
L179 harvested change it please
please proper reference of the slaughtering of the lambs
which GI organs were collected? L180
2.1.4 Visceral mass data.
197 Components of the digestive tract (GIT)????
gastro intestinal tract ..........
statistical analysis which software package and test were used please add
L219 Temperature and relative humidity during the experiment are presented in Table 2. 219 Average minimum and maximum estimated THI were 74.99 and 87.90, respectively [14]. why reference here?
L219-223 revise please
results are totally poor written please revise the results and add the sub sections of the results
discussion section is also poor written it can not be acceptable in this form.
the quality as well as the size of the manuscript is not suitable for research article must be change it from research article to short communication please.
conclusion section must be revised please
check the English please
please check the English language throughout the manuscript
Author Response
Response to REVIEWER 2
AU: We are grateful to reviewers for the time and effort in helping improve the quality of the manuscript. The observations were wise and timely which permit the improvement substantially the manuscript. We have addressed the concerns in our revised manuscript accordingly.
All changes and correction made are highlighted in yellow in the corrected version of the manuscript.
Responses
The conducted a research on the Ionophore Monensin Supplemented in High-Energy Diets for Lambs Long-term Finished Under Subtropical Climate. it looks good but it need improvement before any decision
RW: simple summary is not good please revise.
AU: simple summary was improved as suggested
RW: in abstract please add a line about the background of the study.
AU: a sentence with background information of the study was inserted as is suggested.
RW: how many lambs were selected? and how many lambs were selected in each group/
AU: We selected 48 (from 55), with 12 lambs in each group. This is specified in the corrected version of the manuscript.
RW: please ad the age of the lambs /
AU: Age information was inserted as is suggested.
RW: Revise line L27-31.
AU: paragraph was corrected.
RW: Revise L 31-33L 33-34 how?
AU: Information requested was inserted
RW: Results must be revised in abstract section
AU: Results was revised and improved when required.
RW: Please revise keywords
AU: keywords was corrected
RW: Introduction is poor written and cannot be acceptable in this condition
AU: introduction was improved as is suggested
RW: Give some title to first paragraph of the MM section
AU: Done
RW: please divided in to different sections
AU: Done
RW: animal experiment design
AU: Subheading title was changed as is suggested
RW: sample analysis
AU: Done
RW: change L102-103 please rewrite
AU: Done
RW: animal research design 103-136 must be revised confusing please rewrite and simplify it for proper understanding.
AU: The paragraph contained in L103-136 was rewritten in order to be more understandable its description
RW: how you have prepared the Diet? add the proper reference please
AU: Detailed description how the basal diet was prepared was inserted
RW: please check the diet composition carefully it looks not good revise it accordingly please.
AU: Thanks. Diet composition was corrected
RW: please carefully check the short forms which you have used in the manuscript
AU: Abbreviations and short forms were carefully checked and corrected when required.
RW: 2.3 calculation please revise it
AU: Calculation was revised and corrected when required
RW: L179 harvested change it please
AU: Done
RW: please proper reference of the slaughtering of the lambs
AU: A reference for slaughtering was inserted as is suggested
RW: which GI organs were collected? L180.
AU: Specification of the organs collected was exposed in “visceral mass data” subheading
RW: 2.1.4 Visceral mass data.
AU: Dot at final was removed.
RW:197 Components of the digestive tract (GIT)????
AU: The description was changed as: components of gastrointestinal tract.
RW: Statistical analysis which software package and test were used please add
AU: Statistical software information was inserted as is suggested.
RW: L219 Temperature and relative humidity during the experiment are presented in Table 2. 219 Average minimum and maximum estimated THI were 74.99 and 87.90, respectively [14]. why reference here?
AU: It was a mistake, reference was removed.
RW: L219-223 revise please
AU: The paragraph was revised and corrected
RW: results are totally poor written please revise the results and add the sub sections of the results
AU: Subsection was added and results was improved as is suggested
RW: Discussion section is also poor written it cannot be acceptable in this form.
AU: Honorable reviewer, I am not clear about the observation of "unacceptable form". The manuscript is written following the editorial and style standards of the journal. In the other way, the discussion addresses the central points and is duly supported by scientific arguments. Even so, in order to satisfactorily cover your observation, discussion was expanded in some variables.
AU: the quality as well as the size of the manuscript is not suitable for research article must be change it from research article to short communication please.
RW: I think this observation be directed to the Editor
RW: conclusion section must be revised please
AU: Conclusion was revised and improved
RW: check the English please
AU: English writing was revised by a native speaker and co-author of the manuscript (Dr. Richard A. Zinn, UC Davis)
Reviewer 3 Report
Line 27 and subsequent: The need for two decimal places in LW is excessive when declaring weights above 10 kg, consider expressing to the tenth here and throughout the paper.
Line 62: Change ["Natural"] to Natural and explain your perceived discrepancy with the terminology.
Line 114: Change 'Pens have 6 m2' to 'Pen dimensions were 6m2'
Line 121: Provide justification for 28 mg Monensin treatment. Even if it is industry standard, it should still be stated.
Line 123: Space between vitamin and D3.
Line 127: CRINA plus HyD has not been expressed previously and should be elaborated on or substituted with EO+D3.
Line 147: Because feed offered was either increased or decreased dependent on weight, it would be worth disclosing how often each treatment observed no refusals throughout the experiment. Because the interpretation of your study is heavily dependent on DMI and efficiency, this data is worth stating.
Lines 154-158: These descriptions for CP and aNDFom analysis should be added to Table 1 as a footnote.
Line 163: Elaboration of ADG/DMI should be provided. What DMI was used to make this calculation? Considering ADG is the difference in SBW from d 121 and d 1, was average DMI used? If so, please state.
Line 170: Replace ',' with '.' to begin a new sentence ". Those values were calculated..."
Line 171: 'in the basal diet'
Line 173: You've described a quadratic calculation for 'dietary net energy'; however, you've stated 'x' as being equal to the 'net energy for maintenance'. Please clarify as dietary net energy, which is assumed to be total dietary net energy, is different from net energy for maintenance.
Line 211: Because pen has been stated as the experimental unit for growth performance data, how was gain, efficiency, and DMI assessed? Were they averaged for each pen and used for statistical inference? If lamb was the 'observational unit' and gain, efficiency, and DMI for each lamb was used for statistical inference, lamb within pen would be part of this model and should be stated as such. More elaboration is needed regardless of the approach.
Line 219: Hours spent in a particular THI may be helpful to understand heat load. Providing the average Ta and RH throughout an average day for each EXP week may give better insight on the time spent over a THI threshold. consider adding through supplementary data.
Line 224: Report in Table 1, adding the observed equivalence for each treatment ingredient.
Line 231: Is this greater than or equal to or just equal to?
Line 237: Change 'to' to 'for'
Table 2: Consider adding location of experiment in the footnotes.
Table 2: Temp and RH should be expressed to the tenth, especially if your deviation is also expressed this way.
Line 258: Consider revising 'livestock production represents challenges to sustain efficient productivity' to 'challenges exist when sustaining efficient productivity in livestock systems.'
Line 260: Change consistent to 'that is consistent'.
Line 261: Consider revising sentence structure.
Line 269: Provide clarification here. I do not see this statement represented in your table. Some feedback on how this data was generated is appreciated (i.e. 1.121/1.119 kg DM/d).
Line 274: Table 2 should be Table 3 and all other subsequent tables should be re-labeled.
Table 2b: Superscript letters which highlight statistical differences.
Line 284: Was there an opportunity to use the same EO blend with an additional supplementation of D3 or BS? I understand that these two treatments are product on the market, but it is difficult to isolate the effect of EO, D3, and BS.
Line 290: Change affect to effect
Line 296: change 'may to associated' to 'may be associated'
Table 3: Consider decimal alignment throughout the table.
Table 3: If CCW is not an approved acronym by Animals and MDPI, please define in the footnotes.
Table 4: Decimal align
Line 349: insert 'to' before MON.
Line 361: Limited discussion is provided for Table 4; however, a paragraph is written on the results displayed in Table 4. Please elaborate more on these results by either expanding on the antimicrobial/anti-inflammatory statement made or including context from other studies.
Line 364: Reflecting on the objective of this work, to compare two EO & D3 or BS treatments to monensin, you do well to compare EO & D3 to monensin but state little on EO & BS vs monensin. An expansion of the conclusions with this in mind is warranted.
Overall quality of English is great. There are a few grammatical suggestions in the comments which should be considered. Consider revising some sentences' structure as they may be run-ons or periods may be improperly placed.
Author Response
Response to REVIEWER 3
AU: We are grateful to reviewers for the time and effort in helping improve the quality of the manuscript. The observations were wise and timely which permit the improvement substantially the manuscript. We have addressed the concerns in our revised manuscript accordingly.
All changes and correction made are highlighted in yellow in the corrected version of the manuscript.
Responses
RW: Overall quality of English is great. There are a few grammatical suggestions in the comments which should be considered. Consider revising some sentences' structure as they may be run-ons or periods may be improperly placed.
RW: Line 27 and subsequent: The need for two decimal places in LW is excessive when declaring weights above 10 kg, consider expressing to the tenth here and throughout the paper.
AU: All weight declaration in the document above 10 kg were expressed as is suggested.
RW: Line 62: Change ["Natural"] to Natural and explain your perceived discrepancy with the terminology.
AU: Placing quotes around the natural word was just a typographical error that has already been corrected
RW: Line 114: Change 'Pens have 6 m2' to 'Pen dimensions were 6m2'
AU: Done
RW: Line 121: Provide justification for 28 mg Monensin treatment. Even if it is industry standard, it should still be stated.
AU: Justification was provided as is suggested
RW: Line 123: Space between vitamin and D3.
AU: Correction was done
RW: Line 127: CRINA plus HyD has not been expressed previously and should be elaborated on or substituted with EO+D3.
AU: CRINA plus HyD was substituted with EO+D3 as is suggested
RW: Line 147: Because feed offered was either increased or decreased dependent on weight, it would be worth disclosing how often each treatment observed no refusals throughout the experiment. Because the interpretation of your study is heavily dependent on DMI and efficiency, this data is worth stating.
AU: As is known, in this type of study it is important that the animals are not restricted in their feed consumption. But it is sought that they do not have too much food since it causes the selectivity of the components of the diet. The feed bunk management used was the "licked feed bunk". There is a record of consumption per pen during the adaptation to the experimental diets. When the experiment started, feed delivery was offered approximately 10% above of registered consumption, when the feed bunk appeared "licked" for 2 consecutive days the feed supply was increased by 10%. On the contrary, when feed refusals were above 10% of consumption, feed delivery was reduced 10% until intake became stable. This represent around maximal 5% of refusal/lamb (~50 g/kg).
RW: Lines 154-158: These descriptions for CP and aNDFom analysis should be added to Table 1 as a footnote.
AU: Done
RW: Line 163: Elaboration of ADG/DMI should be provided. What DMI was used to make this calculation? Considering ADG is the difference in SBW from d 121 and d 1, was average DMI used? If so, please state.
AU: Pertinent specification about of average DMI used to calculate feed efficiency was done as is suggested.
RW: Line 170: Replace ',' with '.' to begin a new sentence ". Those values were calculated..."
AU: Done
RW: Line 171: 'in the basal diet'
AU: The correction was made
RW: Line 173: You've described a quadratic calculation for 'dietary net energy'; however, you've stated 'x' as being equal to the 'net energy for maintenance'. Please clarify as dietary net energy, which is assumed to be total dietary net energy, is different from net energy for maintenance.
AU: Thanks! The correction was made.
RW: Line 211: Because pen has been stated as the experimental unit for growth performance data, how was gain, efficiency, and DMI assessed? Were they averaged for each pen and used for statistical inference? If lamb was the 'observational unit' and gain, efficiency, and DMI for each lamb was used for statistical inference, lamb within pen would be part of this model and should be stated as such. More elaboration is needed regardless of the approach.
AU: Thanks! The statement was rewritten as follows: Growth performance data (gain, gain efficiency, and dietary energetics), DM intake, and carcass data were analyzed as a randomized complete block design, with the pen as the experimental unit, using procedures of SAS software [30], with treatment and block as fixed effects and experimental unit within treatment as random effect. Visceral organ mass data was analyzed using the MIXED procedures of SAS software [30], with treatment and pen as fixed effects and interaction treatment × pen and individual carcasses within pen by treatment subclasses as random effects
RW: Line 219: Hours spent in a particular THI may be helpful to understand heat load. Providing the average Ta and RH throughout an average day for each EXP week may give better insight on the time spent over a THI threshold. consider adding through supplementary data.
AU: The following information was inserted as is suggested: Daily maximal THI exceeded 80 for 119 d of the 121 days of the trial, corresponding to “danger” conditions according to code of Mader et al. [26]. The number of hours per day that THI exceeded 80 was 6.2±1.02.
RW: Line 224: Report in Table 1, adding the observed equivalence for each treatment ingredient.
A: Done
RW: Line 231: Is this greater than or equal to or just equal to?
AU: Thanks! Is greater than…Correction was made
RW: Line 237: Change 'to' to 'for'
AU: Change was made as suggested
RW: Table 2: Consider adding location of experiment in the footnotes.
AU: location was added as suggested
RW: Table 2: Temp and RH should be expressed to the tenth, especially if your deviation is also expressed this way.
AU: Change was done as is suggested
RW: Line 258: Consider revising 'livestock production represents challenges to sustain efficient productivity' to 'challenges exist when sustaining efficient productivity in livestock systems.'
AU: Good point! The statement was changed following your suggestion.
RW: Line 260: Change consistent to 'that is consistent'.
AU: Done
RW: Line 261: Consider revising sentence structure.
AU: Sentence was restructured
RW: Line 269: Provide clarification here. I do not see this statement represented in your table. Some feedback on how this data was generated is appreciated (i.e. 1.121/1.119 kg DM/d).
AU: Thanks! The ratio was corrected as: (1.123/1.119 kg DM/d). where 1.123 is the average DMI for EO+BS lambs (Table 3) and 1.119 is expected DMI according to NRC 2007 equation approach (indicated previously).
RW: Line 274: Table 2 should be Table 3 and all other subsequent tables should be re-labeled.
AU: Now the Tables are numbered correctly
RW: Table 2b: Superscript letters which highlight statistical differences.
AU: Done
RW: Line 284: Was there an opportunity to use the same EO blend with an additional supplementation of D3 or BS? I understand that these two treatments are product on the market, but it is difficult to isolate the effect of EO, D3, and BS.
AU: Honorable reviewer, not in this experiment. But our group have been the opportunity to test supplementation alone of EO and BS and EO or BS plus other feed additives in other experiments (some already published).
RW: Line 290: Change affect to effect
AU: Done
RW: Line 296: change 'may to associated' to 'may be associated'
AU: Done
RW: Table 3: Consider decimal alignment throughout the table.
AU: Done
RW: Table 3: If CCW is not an approved acronym by Animals and MDPI, please define in the footnotes.
AU: CCW is defined in Mat and Met section (subheading 2.6)
RW: Table 4: Decimal align
AU: Done
RW: Line 349: insert 'to' before MON.
AU: Done
RW: Line 361: Limited discussion is provided for Table 4; however, a paragraph is written on the results displayed in Table 4. Please elaborate more on these results by either expanding on the antimicrobial/anti-inflammatory statement made or including context from other studies.
AU: We expand the discussion as is suggested
RW: Line 364: Reflecting on the objective of this work, to compare two EO & D3 or BS treatments to monensin, you do well to compare EO & D3 to monensin but state little on EO & BS vs monensin. An expansion of the conclusions with this in mind is warranted.
AU: Conclusion was rewritten in order to cover your wise suggestion
Round 2
Reviewer 2 Report
The authors have revised the manuscript as per suggestions but still need some improvement.
the title looks vague please revise
the summary of the paper has been revised.
plz revise L32-33
L-58-61 revise please
L 66 add reference please DOI: 10.29261/pakvetj/2023.002 https://doi.org/10.3389/fvets.2023.1145610
revise these sub-title please 2.1 Location in which the study was performed
2.2 Weather measurement and temperature humidity index (THI) estimation
3.2 Relative intake of additives
3.3 Growth performance and dietary energetics
why the results tables are added in discussion section ?????
please adjust all results tables in results section
in conclusion section please add the future recommendations
please delete the old and unnecesary references. you have added more than 60 references please revise and delete an appropriate references
English language must be revised please still many corrections
n/a
Author Response
AU: We are grateful to reviewer for the time and effort in helping improve the quality of the manuscript. The observations were wise and timely which permit the improvement substantially the manuscript. We have addressed the concerns in our revised manuscript accordingly.
All changes and correction made are highlighted in yellow in the corrected version of the manuscript.
Responses
RW: The title looks vague please revise
AU: Following your suggestion, the Title was rewritten in order to be more specific
RW: the summary of the paper has been revised.
AU: ok
RW: plz revise L32-33
AU: Sentence was revised and corrected
RW: L-58-61 revise please
AU: AU: Sentence was revised and corrected
RW: L 66 add reference please DOI: 0.29261/pakvetj/2023.002 https://doi.org/10.3389/fvets.2023.1145610
AU: We add only the second reference, the first reference is more directed to heat stress and mammary gland metabolism
RW: Revise these sub-title please 2.1 Location in which the study was performed
AU: Sub-title 2.1 was revised and corrected
RW: 2.2 Weather measurement and temperature humidity index (THI) estimation
AU: Sub-title 2.2 was revised and corrected
RW: 3.2 Relative intake of additives
AU: Sub-title was changed as: “Additives intake”
RW: 3.3 Growth performance and dietary energetics
AU: Sub-title was changed as: “Growth performance and dietary energy”
RW: why the results tables are added in discussion section ?????
AU: Tables were relocated
RW: please adjust all results tables in results section
AU: Tables were relocated
RW: in conclusion section please add the future recommendations.
AU: Following your suggestion we added some recommendations
RW: Please delete the old and unnecesary references. you have added more than 60 references please revise and delete an appropriate reference.
AU: Honorable reviewer, 80% of old references are related to monensin studies. As you well know, most of the information on monensin was generated decades ago. All the cited studies are relevant and necessary to support the arguments discussed. On the other hand, the other old references are from sources that support methodologies and procedures described in materials and methods, therefore they are also necessary to be cited.
RW: English language must be revised please still many corrections
AU: All manuscript was exhaustively revised and corrected when it was necessary